# Turing's Conceptual Engineering

Marcin Miłkowski

Institute of Philosophy and Sociology, Polish Academy of Sciences, ul. Nowy Świat 72, 00-330 Warszawa, Poland; mmilkows@ifispan.edu.pl

**Abstract:** Alan Turing's influence on subsequent research in artificial intelligence is undeniable. His proposed test for intelligence remains influential. In this paper, I propose to analyze his conception of intelligence by relying on traditional close reading and language technology. The Turing test is interpreted as an instance of conceptual engineering that rejects the role of the previous linguistic usage, but appeals to intuition pumps instead. Even though many conceive his proposal as a prime case of operationalism, it is more plausibly viewed as a stepping stone toward a future theoretical construal of intelligence in mechanical terms. To complete this picture, his own conceptual network is analyzed through the lens of distributional semantics over the corpus of his written work. As it turns out, Turing's conceptual engineering of the notion of intelligence is indeed quite similar to providing a precising definition with the aim of revising the usage of the concept. However, that is not its ultimate aim: Turing is after a rich theoretical understanding of thinking in mechanical, i.e., computational, terms.

**Keywords:** Turing test; conceptual engineering; digital humanities; artificial intelligence

## 1. Introduction

Alan Turing's influence on the development and progress of computer science and artificial intelligence cannot be overestimated. His accomplishments in these areas remain also striking in regard to general philosophical issues, such as the very possibility of artificial minds. Simply, Turing's proposal defined the field of inquiry. Despite numerous objections, the Turing test remains the most influential way to consider the possibility of artificial intelligence.

In what follows, my focus is on the foundational 1950 paper that presents the famous test. My goal is to understand Turing's proposal as an exercise of conceptual engineering aimed at the theoretical understanding of thinking. Conceptual engineering is the project of revising, or ameliorating, our concepts, rather than analyzing them [1,2]. My interpretation of the test is complemented by a more "distant reading" of Turing's oeuvre: by studying general patterns of his language use, parts of his conceptual framework are sketched. To accomplish this aim, a corpus of Turing's writings was compiled and then analyzed computationally.

Section 2 focuses on a close reading of Turing's approach to the issue of thinking machines. It is argued that Turing proposed an alternative to conceptual analysis: He engineered the notion of intelligence through a thought experiment. However, the aim of the experiment in his engineering remains somewhat debatable. I argue that it aims at revising our notion of thinking by substituting it with a notion of intelligence. However, the traditional methods of interpretation do not suffice to ascertain why the notion of intelligence was substituted by Turing for the notion of mind or thinking. It remains also unclear whether the Turing test provides an operational definition of intelligence. This leads to a general discussion of the semantics of crucial psychological terms in Turing's corpus in Section 3.

In Section 3, I introduce the methods of distant reading, and by analyzing Turing's writings with language technology, I aim to provide tentative answers to two questions:

(1) Why did Turing rely on the notion of intelligence, rather than that of the mind? (2) What role does the Turing test play: is it an operational definition or not? As it turns out, at least some open questions about Turing's engineering can be addressed by applying the methods of distant reading.

## 2. The Turing Test as an Intuition Pump

In this section, I will first interpret the Turing test as a thought experiment for revising our notion of intelligence. Then, I will contrast this experiment with another instance of conceptual engineering, found in his foundational paper on computation [3]. Finally, I discuss two questions that remain undecided by traditional methods of philosophical interpretation, which can be dubbed "close reading".

Turing's immensely influential paper on computational intelligence starts with a discussion of an alternative approach to the question of whether machines can think:

> I propose to consider the question, 'Can machines think?' This should begin with definitions of the meaning of the terms 'machine' and 'think'. The definitions might be framed so as to reflect so far as possible the normal use of the words, but this attitude is dangerous. If the meaning of the words 'machine' and 'think' are to be found by examining how they are commonly used it is difficult to escape the conclusion that the meaning and the answer to the question, 'Can machines think?' is to be sought in a statistical survey such as a Gallup poll. But this is absurd. Instead of attempting such a definition I shall replace the question by another, which is closely related to it and is expressed in relatively unambiguous words. ([4], p. 433)

In this passage, Turing's argument seems to be as follows:

1. Answering questions such as "Can machines think?" involves providing lexical definitions of the meaning of component terms.
2. Lexical definitions of the meaning of component terms reflect the normal use of the words.
3. If lexical definitions reflect normal use, then answering questions such as "Can machines think?" should rely on statistical surveys such as Gallup polls.
4. Therefore, answering questions such as "Can machines think?" should rely on statistical surveys.
5. However, answering questions such as "Can machines think?" should not rely on statistical surveys.

Thus, answering questions such as "Can machines think?" does not involve providing lexical definitions.

This is a valid reductio. Lines 2 and 3 imply line 4 via modus ponens. Line 4, however, leads to a contradiction with an additional premise 5, which is accepted by Turing in passing without much discussion. He only states that relying on statistical surveys in this regard would be "dangerous". Note that contemporary experimental philosophers who rely on statistical surveys and similar empirical methods might also explicitly embrace a premise that implies line 4 or something similar. It might also seem that if Turing's assumption is that the study of lexical definitions is not useful for answering conceptual questions, then also, by the same token, the computational methods used later in this paper are equally dangerous. Nonetheless, to avoid contradiction between lines 4 and 5, Turing apparently suggests that we should simply drop the premise stated on line 1. This then is his indirect argument against relying on describing the meaning of concepts, which eventually boils down to providing lexical definitions, in answering questions such as "Can machines think?".

What would be the danger of relying on statistical surveys to answer such philosophical questions? Plausibly, Turing, along with philosophers of the ordinary language of his time, was well aware of the fact that usage patterns change over time, and that "reading off" philosophical views from particular linguistic usage is akin to committing a simple

fallacy of confusing "is" with "ought".[1] A description of a usage pattern in language does not suffice to justify the general claim that this pattern should be followed, even if uttered in suitable contexts and taken for face value. For example, from the fact that there are many (fossilized) expressions such as "The sun sets", which might be taken, at face value, to express a commitment to a geocentric theory of universe, it does not follow that the theory is widely assumed to be true or endorsed by a competent speaker uttering such an expression. At the same time, common usage patterns are pieces of evidence used by linguists to formulate prescriptive rules; when prescriptive rules clash with usage, in particular educated usage, they come under heavy criticism (see, e.g., [7]). Nonetheless, educated usage is infused with prescriptive attitudes that language users take in their language production. For Turing, educated usage is, however, irrelevant to the truth of philosophical questions, even if it could decide issues of spelling or grammar.

When contrasting his own proposal with the question 'Can machines think?', which is "too meaningless to deserve discussion", he stresses that it is "expressed in relatively unambiguous terms" ([4], p. 433). This suggests that Turing considers the term "thinking" too ambiguous for discussing the issue at hand. In this, he explicitly rejects what was voiced by Wittgenstein in a widely quoted passage: "The trouble is rather that the sentence, 'A machine thinks (perceives, wishes)' seems somehow nonsensical. It is as though we had asked 'Has the number 3 a colour?'" ([8], p. 47). Wittgenstein and his followers consider the question to involve a category mistake. By answering the question, or even replacing it with another, Turing opposes Wittgenstein's view. In a study of Wittgenstein's remarks on artificial intelligence, Shanker suggests that "it seems plausible to suppose that the Turing Test represents Turing's opposition to Wittgenstein's critique, using a Wittgenstein-like argument" ([9], p. 2). The aim of Turing's paper is to eventually revise the notion of thinking so that the question would be not only meaningful but also possibly answered in the positive. What is striking is that, instead of analyzing extant meaning, Turing proceeds to justify a significant conceptual change through his test.

Turing's argument against relying on statistical surveys is but one in an ongoing debate between descriptivists and revisionists about philosophical concepts. Instead of understanding and analyzing our existing conceptual machinery, revisionists propose revising them. At the turn of the 21st century, Simon Blackburn dubbed the project of assessing and ameliorating our concepts "conceptual engineering" [1]. These revisions involve various changes, from sharpening, through explication and partial replacement, to elimination. What Turing does is to offer a partial replacement of the notion of thinking by a sharpened notion of intelligence, which is justified through his test.

The prominent counterargument against introducing conceptual revisions is that they change the topic of inquiry [8]. Turing anticipated that his proposal could be similarly contested, conceding that he "cannot altogether abandon the original form of the problem, for opinions will differ as to the appropriateness of the substitution" ([4], p. 442). There are two features of the Turing test that deserve our attention in this context. First, it relies on common-sense judgement (or intuitive assessment), both of the participants of the test and of the readers of his imaginary scenario. The answer to the question is not decided by fiat in a stipulative formal manner, which he considered generally implausible.[2] Second, the scenario of the test must retain at least some resemblance with the original question to be appropriate, which implies that Turing believed that his test did not deviate from the *topic* of machine thinking. Defenders of conceptual engineering argue that the sameness of topics of inquiry is more coarse-grained than the sameness of particular concepts [2]. In analyzing the test, we should then see what it is that it seems apt for Turing as a replacement of the original question.

Two questions stand out in this context. Why is it appropriate to replace the notion of thinking with the notion of intelligence, as exhibited in the test? And why did Turing provide a test rather than an explicit definition? A number of interpretations suggest that its main function is to provide an operational definition of "intelligence", which is used

as a proxy for "thinking" [11,12]. However, before I address these questions, let me now discuss the test and then relate it to its purposes in Turing's conceptual engineering.

Here, my presentation of the Turing test will be kept brief (for extended discussions, see [13–17]). In short, what Descartes thought to be unavailable for any sort of mechanism, viz., communication in natural language, is considered by Turing to be the hallmark of intelligence. In *Discourse on the Method*, Descartes contended that no mechanism could ever "use words, or put together other signs, as we do in order to declare our thoughts to others" ([18], p. 140). Turning the tables, Turing considers natural language use to be sufficient for intelligence. The core idea is that there could be sufficient similarity between how machines and human beings use words. The possibility that there be such similarity is the point of his version of the imitation game, in which a judge is asked to determine whether they converse with a machine or a human being.

Thus, the goal of the imitation game is to test whether a machine might imitate human conversional ability to the extent that it would be indistinguishable from a human being. Being indistinguishable puts the machine into the same class of abstraction, showing that it is equivalent to the human being under this respect. This equivalence concerns only observable behavior. The machine need not provide the same answers, given the same question as a human being. Verbal behavior need not be produced in any way equivalent to how human beings produce it.[3] In other words, the equivalence in question is far less demanding than the ones implied in the subsequent simulation research in psychology [21].

Turing's version of the imitation game, adapted from a party game, serves a function of what Dennett calls an "intuition pump" [22]: It provides a thought experiment to engineer the concept of intelligence as applied to machines. While intuition pumps need not serve revisionary roles, they can also be used for revising or even rejecting concepts. Following Dennett, one may analyze Turing's test as a device for "pumping" intuitions that depend on the "pump" settings.

The first set of "pump" settings ensures that the machine remains in disguise for a judge by requiring that the communication be purely textual, without the presence of biological bodies, with a proper speed of communication, that the machine be not obliged to declare its true identity, etc. Some of these settings could be varied without influencing the result of the test if computational technologies would allow the machine, for example, to speak English with no discernible "machine accent" and parse spoken English. The essential thing is that the playing ground be even for comparing human beings and machines.

The second set of settings is related to the contents of the conversation: It should not be constrained or limited, which could make the test too easy. Moreover, the comparison should be made between the machine and a typical adult speaker. The success of Parry in simulating a paranoid patient [23] did not constitute a milestone in the development of general artificial intelligence.

Finally, the third set of settings is related to the judges, who should focus on the conversation alone.

By analyzing the construction of the intuition pump, we can discover that the test makes the notion of machine intelligence relational. As Proudfoot observes, Turing embraced an "externalist" conception of intelligence: "whether or not machines think is in part determined by social environment, in the form of the interrogator's responses" [13]. This is a double-edged sword.

On the one hand, this externalism may undermine the intersubjective agreement over a given instance of a test. Judges may disagree. Indeed, substantial disagreement is typical [24]. This could be the reason why, during a radio debate in 1952, Turing proposed that a jury should decide: "A considerable proportion of a jury, who should not be expert about machines, must be taken in by the pretence" ([15], p. 495). This indicates that Turing did not want any preconceptions about machines and computation to take precedence over the conversational dynamics (so this constrains how judges should be selected). As Colby observed, in the original imitation game, whose aim was to distinguish a man mimicking

a woman from a real woman, there can be no real experts either, because "there exist no known experts for making judgments along a dimension of womanliness" [23]. However, this is not a bug but a feature for Turing.

If we were to treat this as a specification of an experimental protocol, then there is a considerable similarity of the roles of the members of the jury and human participants who are asked to a classify a certain entity. In contemporary psychology or language technology, human participants are often asked to provide judgments that are typically limited to a forced choice among several options. These judgments are used in coding the structure ascribed to videos, images, gestures, written or spoken linguistic expressions, etc. Currently, crowd-sourced annotation tasks are usually used to produce datasets for machine learning purposes. In contrast, however, to the Turing test, the choices made by human annotators are usually constrained theoretically by detailed instructions provided by experts. This means that experts are actually responsible for designing such instructions, while judgments are made independently, usually by at least a pair of human beings working independently (to avoid amplifying their individual biases), and then made consistent in various ways. For example, one way to resolve the disagreement is to appeal to a supervising annotator. Another is to hold a discussion between annotators until they resolve the disagreement themselves. None of this is proposed for the Turing test—this is partly because the judgments are not used downstream for further processing. However, this is also because the test is supposed to demonstrate that the machine under the test has the conversational capacity that corresponds to a common-sense judgment. After all, one should not expect intuition pumps to provide precise and generalizable performance metrics.

However, there is a price to pay: passing the test, in contrast to other tasks in language technology, is not a tangible goal, because it is virtually impossible to break down the task into its components and provide a metric of how good a given model performs. While many language-related capacities, such as translation, are difficult to evaluate automatically, e.g., because various human translators might produce vastly different translations given the same original input document, there are some measures that could be used to see whether there is some progress or not (see, e.g., [25] for a discussion of the BLEU score). Not so for the Turing test. This explains why the Turing test has remained outside the purview of mainstream language technology.[4]

On the other hand, Turing's approach seems to align well with the conceptual changes inspired by the subsequent development of artificial intelligence. As many have noticed, we tend to consider machine performance less intelligent when we become acquainted with it ([26], p. 204). Douglas Hofstadter credited Larry Tessler as the author of the theorem that expressed this effect succinctly: "AI is whatever hasn't been done yet" ([27], p. 601). The modern notion of artificial intelligence seems indeed to be "externalist", or "emotional" ([15], p. 431), as Turing would have it:

> The extent to which we regard something as behaving in an intelligent manner is determined as much by our own state of mind and training as by the properties of the object under consideration. If we are able to explain and predict its behaviour or if there seems to be little underlying plan, we have little temptation to imagine intelligence. With the same object therefore it is possible that one man would consider it as intelligent and another would not; the second man would have found out the rules of its behaviour. ([15], p. 431)

However, just because people consider themselves experts on the machines or algorithms they frequently use (such as the web search engine, which is arguably an instance of fairly complex artificial intelligence), they no longer consider them particularly intelligent. While this is a source of frustration among developers of artificial intelligence, Turing's original position seems to be: just bite the bullet.

Thus, this conception of intelligence in computing machinery is not only highly interactive and externalist, but also prone to changes over time, similar to any other

common-sense conception. Nonetheless, making such changes possible seems to be the goal of this instance of Turing's conceptual engineering.

As an intuition pump, the Turing test is designed to help us to revise the notion of intelligence to make it applicable to machine behavior, at least in selected circumstances, just because of the features of machine behavior. While Turing has been repeatedly described as offering an operationalist or behaviorist definition of thinking [11,12], Jack Copeland argues that "[t]here is no textual evidence to support this interpretation of Turing" ([15], p. 435). Indeed, Turing explicitly said that he did not want to "give a definition of thinking" ([15], p. 494). If his conceptual engineering is not about definitions, operational or not, then what is the purpose of the test?

The test puts a machine before an extremely difficult cognitive task. The difficulty lies in the fact that it is unconstrained, unlike the tasks that Turing considered in 1948, such as games, language learning, translation, cryptography, and mathematics ([15], p. 420). An unbounded human conversation may include any of these topics and more. While a casual conversation is not supposed to provide deep insights into the game of chess or translating Ukrainian poetry, one can discuss these and expect that one can educate one's interlocutor in any subject. Conversation is therefore not merely a linguistic task (as translation), but a task that requires significant cognitive flexibility. A machine that passes a test would disprove Descartes's and Wittgenstein's insistence that (flexible) thinking could not be understood in purely mechanistic terms. That is the ultimate purpose of the test.

However, to rebut Descartes and Wittgenstein, a machine need not actually pass the test. The test works two ways. On the one hand, it is a thought experiment, or an intuition pump. At least some readers of the paper, merely by relying on their common sense, and not on any expert judgement, can imagine a machine successfully passing the test. As such, this act of their imagination demonstrates that machine intelligence or machine thinking is conceivable for them. The test need not be even performed in real life. Instead, it offers a tool for thinking: It opens up a conceptual possibility that machines could display intelligence, contra Descartes and Wittgenstein. This is exactly why it should be considered predominantly an intuition pump: It should lead to the revision of the previously entrenched conviction that thinking cannot be mechanized. The test is supposed to be a proof of concept that machines could display intelligence by "declaring their thoughts". It plays this role even without any machine being able to pass the test. As a consequence, the test is not supposed to provide any quality metric for intelligent machine systems.

On the other hand, by studying what it takes to actually pass the test and performing the test in reality, one can learn how cognitively complex human conversation is and work towards flexible machine intelligence. The bar for success in the conceivability test is fairly low, while it seems to be barely reachable for the actual test, in spite of some progress in language technology tasks such as question–answering.

Yet, Turing tends to conflate both ways the test works. While his paper starts with the conceivability question "Can machines think?", in a radio discussion, he says the test replaces the question "Do machines think?" ([15], p. 495). In fact, however, it is the closing of the gap between the two questions that is the task for Turing's research program on machine intelligence.

To see why the definition is not what Turing is after and what the function of the test is, it is instructive to contrast the Turing test with his proposal to understand computability in terms of universal machines [3]. "Computability" used to be an informal concept, sharpened thanks to the new mathematical notion of machine that formalized it. In this case, Turing proposed a definition to formalize the original notion of computation. He justified the introduction of the new notion of computability with his simple thought experiments that described what a human computer could do with paper and pencil, finite number of instructions, etc. Thus, Turing also relied on intuition pumps in his 1937 paper, but his goal was different. He wanted to defend a particular innovative formalization.

Intuition pumps in his conceptual engineering of "computation" can be analyzed as part of Turing's project of explication of this notion [28,29]. By "explication", Carnap meant "the transformation of an inexact, prescientific concept, the *explicandum*, into a new exact concept, the *explicatum*" ([30], p. 3). Explication aims at replacing older concepts with new ones. In his recent analysis of the explication of the notion of "computability", de Benedetto shows that it proceeded in three steps: the *explicandum* is first replaced with a "semi-formal sharpening of the clarified explanandum", and only after this step the *explicatum* is proposed [29]. Turing proposed to sharpen the clarified notion of effective calculability. As de Benedetto argues:

> This is not a conceptual analysis, this is conceptual engineering of an intuitive notion into, first, a clarified notion and then into a sharpened one. Turing merged these two steps in his famous informal exposition . . . he fixed the use and the context of his explication of effective calculability, abstracting the notion of effective calculability from practical limitations, ruling out infinity and any kind of ingenuity, and focusing on the symbolic processes underneath any calculation process. This disambiguation and clarification of effective calculability belongs to the explication step of the clarification of the explicandum. At the same time, Turing sharpened this clarified explicandum by arguing that what is effectively calculable has to be computable by an abstract human calculator respecting in his actions the bounds that we are now all familiar with, thereby implicitly giving a semi-formal definition of the notion of computorability. This implicit semi-formal axiomatization of effective calculability in terms of actions of a computor belongs instead to the mid-level step of the sharpening of the clarified explicandum. ([29], p. 21)

"Intelligence" or "thinking" are informal concepts, such as "computability" before 1937, but Turing did not propose to formalize them. One could hypothesize that in contrast to "computability" in the 1930s, which was successfully formalized in several distinct but mathematically equivalent ways thanks to the confluence of ideas [31], "intelligence" in the 1940s or 1950s did not even refer to possible capacities of computing machines. Many, such as Wittgenstein, deemed the question of whether machines could think as nonsensical. In other words, while Turing's formalization "computability" was relatively uncontested, his approach to mechanization of thought was not. The role of the test, which is indeed a "Wittgenstein-like argument" ([9], p. 2), must have been different because there was no common-sense agreement that it could even make sense to say that machines think. When stating that he hopes that "at the end of the century the use of words and general educated opinion will have altered so much that one will be able to speak of machines thinking without expecting to be contradicted" ([15], p. 449), Turing reveals the role of his work. It is to clarify the notion of thinking and intelligence so that it could be further studied and mechanized.

Thus, Turing would have to propose a theoretical stipulative definition of "thinking" or "intelligence" in formal terms in addition to his intuition pump, were he to follow his previous footsteps. However, he was not ready to offer a formal definition and a novel, complete mathematical framework. He had genuinely new insights into the role of learning, randomness, organization, and situatedness of intelligent systems in their social environments, which preceded developments in computer science sometimes by many decades, but these did not constitute a comprehensive theory of intelligence. In summary, the role of thought experiments related to what might be accomplished with paper and pencil in his 1937 paper diverged from what was supposed to be accomplished in his 1950 paper: conceptual engineering was aimed at formalization in the first case, while it played a more preliminary motivating role in the latter. What the Turing test does is only to legitimize further clarification of the notion of intelligence as mechanizable in principle. Not only he could not fully explicate the notion, but also a semi-formal description of what exactly makes conversation an instance of intelligence was lacking.

Nonetheless, two features of Turing's engineering remain unclear in light of what has already been said. Turing did not explicitly justify why the notion of "intelligence" would be an appropriate replacement for "thought", "thinking" or "the mind". One could rationally reconstruct his possible justification by studying the use of these terms in British philosophy, cybernetics, and everyday debates after the war. For example, Gilbert Ryle stressed the notion of "intelligence" (or "intelligent" behavior) in his 1949 book [6]. However, in contrast to Turing, who was based in Cambridge, Ryle worked at Oxford, and was not a member of the Ratio club, which was a hub of and springboard for the growing cybernetic movement in the UK. There is some evidence that prominent proponents of cybernetics in the UK, such as W. Grey Walter or Donald MacKay, relied on the notion of "intelligent action" rather than "thinking" as well already in the late 1940s [32,33]. However, the latter one also wrote a paper that answered the very same question that the Turing test was designed for [34]. MacKay framed them as a series of interrelated questions, such as the following:

1. Can an artefact be made to show the behavioural characteristics of an organism?
2. How closely in principle could the behaviour of such an artificial organism parallel that of a human mind? ([34], p. 105).

However, the title of MacKay's paper mentions "mindlike behavior", not "intelligence". Thus, it remains unclear why Turing relied on the notion of "intelligence" rather than "mindlike behavior" or "thinking" in the first place. As Copeland notices, while Turing said that the question "Can machines think" is "meaningless to deserve discussion", he "certainly did not allow this view to prevent him from indulging rather often in such discussion" ([15], p. 491). Why go the roundabout way through intelligence and not talk of thinking outright? Turing's writings seem mildly inconsistent in this respect.

It is also unclear why Turing did not think his test would provide an operational definition, specifying the sufficient condition for intelligence. He could have understood it as a working definition to be adjusted in the future. While his denial of the intention to define the notion is univocal, a number of readers were strongly impressed that his paper actually provides a definition, and an operational one. Operational definitions, even if operationalism as a philosophical doctrine about theoretical terms is unviable, do provide clear pointers to other researchers taking up experimental issues. Why not treat the imitation game test as an operational definition, after all?

To address these two issues, which are not decisively answered by close reading, we shall turn to language technology.

## 3. From Close to Distant Reading of Turing

In this section, I study the interpretational questions about the Turing test from the perspective of language technology. The traditional way to engage intellectually with philosophical arguments is to meticulously reconstruct their historical context and conceptually analyze their contents. This approach has become deeply entrenched in philosophy and clearly has merit. It will be complemented here with a more "distant" way of reading. The notion of distant reading was introduced by Moretti [35] in his discussion of the problems with generalizability of claims in comparative literature. It is humanly impossible to study world literature by reading all published works, but digital humanities provide quantitative tools instead. Additionally, distant reading addresses the issue of how to justify interpretive generalizations by providing intersubjectively accessible and quantifiable means to study textual evidence [36]. The problem at stake is that traditional humanities, including the history of philosophy, is replete with general claims about contents of literary, scholarly, or philosophical work, while its methodology rests on the analysis of particular textual passages. However, these passages are more often than not insufficient to back up such claims. Turing, for example, said that the question "Can machines think?" is meaningless, and then proceeded to discuss it. So, what was his view, after all? Interpretational generalizations derived from distant reading remain risky, but at least they are related to the

textual evidence in an intersubjectively replicable fashion. Thus, my aim is to provide some implicit, but replicable, textual evidence for my interpretation.

Distant reading alleviates the worry that passages are selected in an arbitrary fashion. It is not meant to replace close reading; instead, it can provide additional context to understand the overall project in which Turing was engaged. While distant reading remains outside the mainstream of the history of philosophy, it can address the problems of interpretive bias that plague humanities. For example, in his recent book, Marc Alfano applied distant reading methods to analyze the conceptual apparatus of Nietzsche's moral psychology. His results show that the traditional focus on such notions as "will to power" or "resentment" is largely arbitrary as these notions are not the most prominent in Nietzsche's moral psychology. This section has a similar aim: rely on tools of digital humanities to understand Turing better and provide evidence for textual interpretations in a replicable fashion.

Over the past few years, huge progress has been made not only in language technology, but also in digital humanities, which, to a large extent, rely on this technology. New developments in deep neural networks led to a breakthrough in the performance of multiple applications of language technology, from machine translation, through speech processing and question answering, to text analysis [37]. While there are multiple methods one could use to gain new perspectives on historical work through text analysis and text mining, such as topic modeling or diachronic study of the frequency of certain terms, in this paper, distributional semantics is the approach of choice. According to distributional semantics, the meaning of an expression is determined by what accompanies it in a large body of text.

This approach to semantics has recently dominated in language technology, in particular thanks to its use in successful language models, such as word2vec [38] and Glove [39]. Its roots historically stretch back, however, much more deeply. Already in the 1950s, John Firth advocated for the principle "You shall know a word by a company it keeps!" ([40], p. 179). His idea was that lexicographers present the meaning of words through excerpts, or examples of word collocations. For the purposes of this paper, a collocation is a sequence of words that occur together more often than would be expected by chance alone (for a recent review, see [41]). This is because, he stressed, meaning is indicated by habitual collocations that correspond to common patterns of usage.

These patterns are what sometimes remains implicit in the body of text and can be easily missed. Close reading is indeed indispensable in following Turing's line of argument, but the insight into more general usage patterns can provide more context. This is what "distant reading" is about: finding the overall semantic structure.

Let us now turn to this structure. The written oeuvre of Alan Turing remains rather small. For the purposes of the current study, a corpus of his published and unpublished work was compiled, including all his work included in [15] but also additional work contained in [42] (i.e., "Proposal for Development in the Mathematics Division of an Automatic Computing Engine (ACE)", "Lecture to the London Mathematical Society on 20 February", "Checking a large routine") as well as unpublished work on morphogenesis from [43] and some parts of transcripts on Turing's Enigma book.[5] The resulting corpus has 227,757 tokens (185,052 words).

For the purposes of semantic analysis, the corpus was processed using SketchEngine (www.sketchengine.eu, accessed on 10 June 2022), which is a state-of-the-art corpus analysis software [46]. One interesting feature of SketchEngine is the ability to extract the similarity of meanings of terms in a corpus, based on distributional relations inherent in it. They can be visualized using diagrams (see below Figures 1–4, 7 and 8), which contain the term of interest in the center, while the distance from the center depicts the degree of semantic similarity.[6]

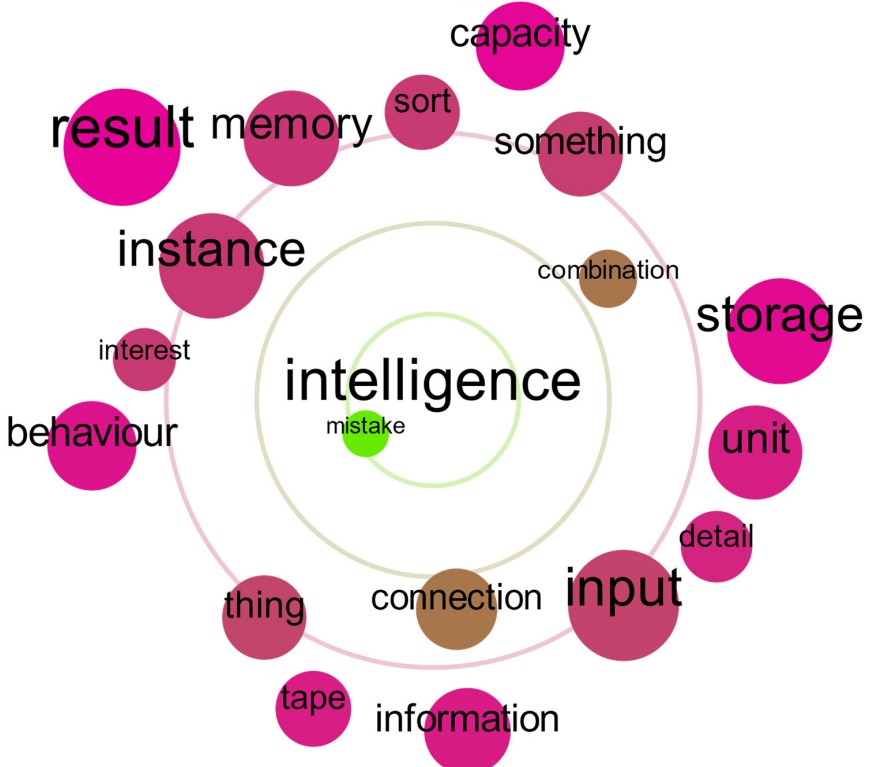

**Figure 1.** Terms semantically close to "intelligence". Font and circle sizes correspond to the term frequency in the corpus, while distances reflect semantic similarity.

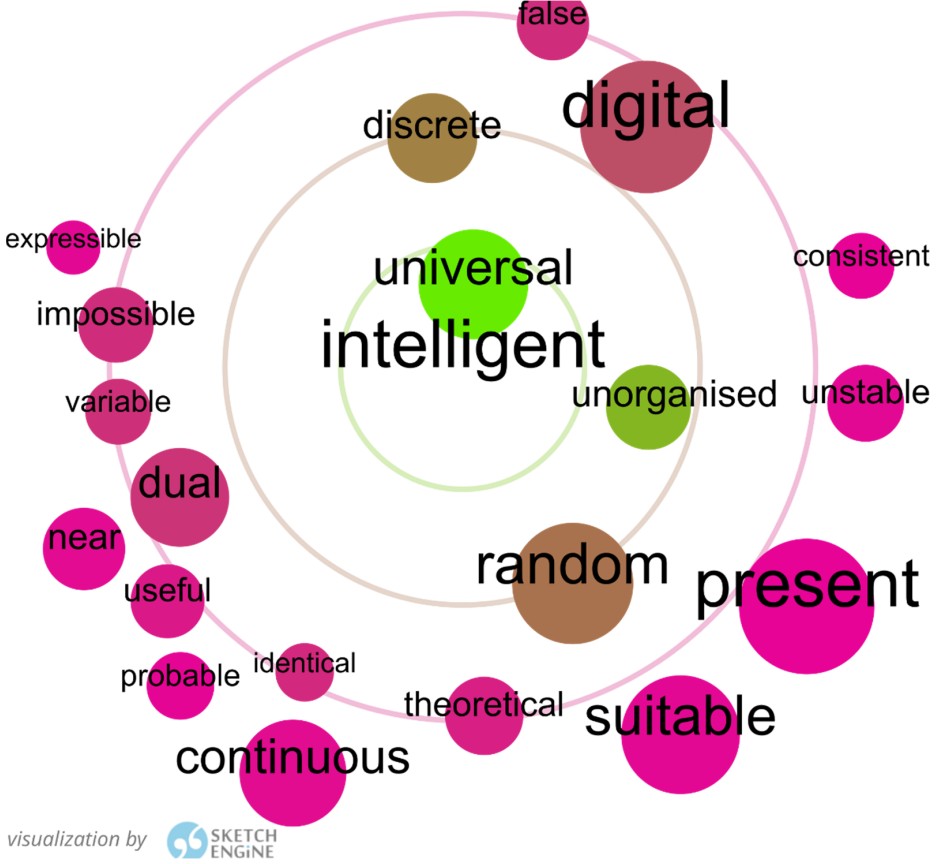

**Figure 2.** The visualization of terms semantically similar to "intelligent" in Turing's writings.

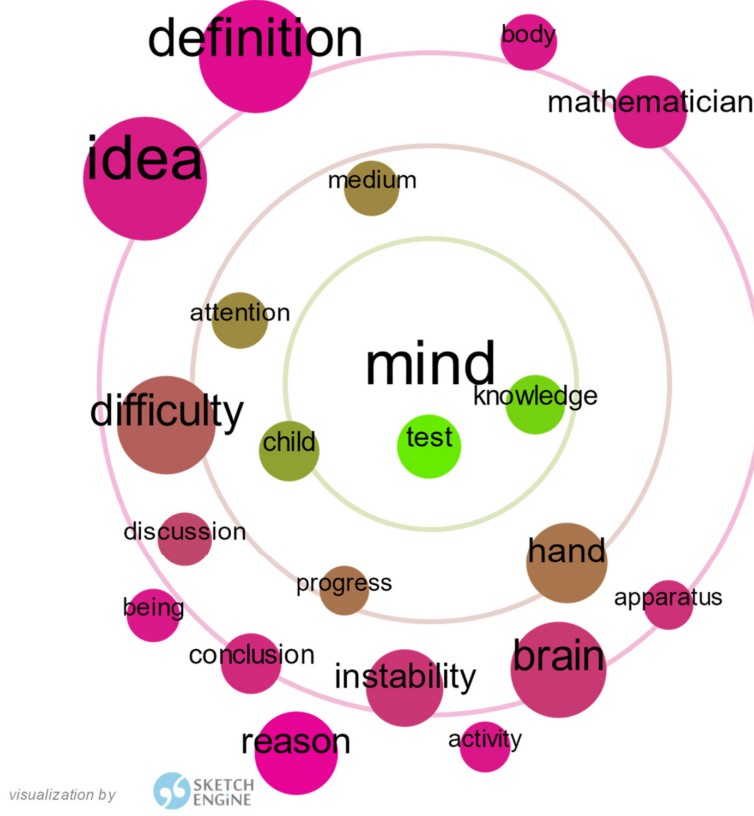

**Figure 3.** This diagram depicts terms semantically close to the term "mind" in Turing's corpus.

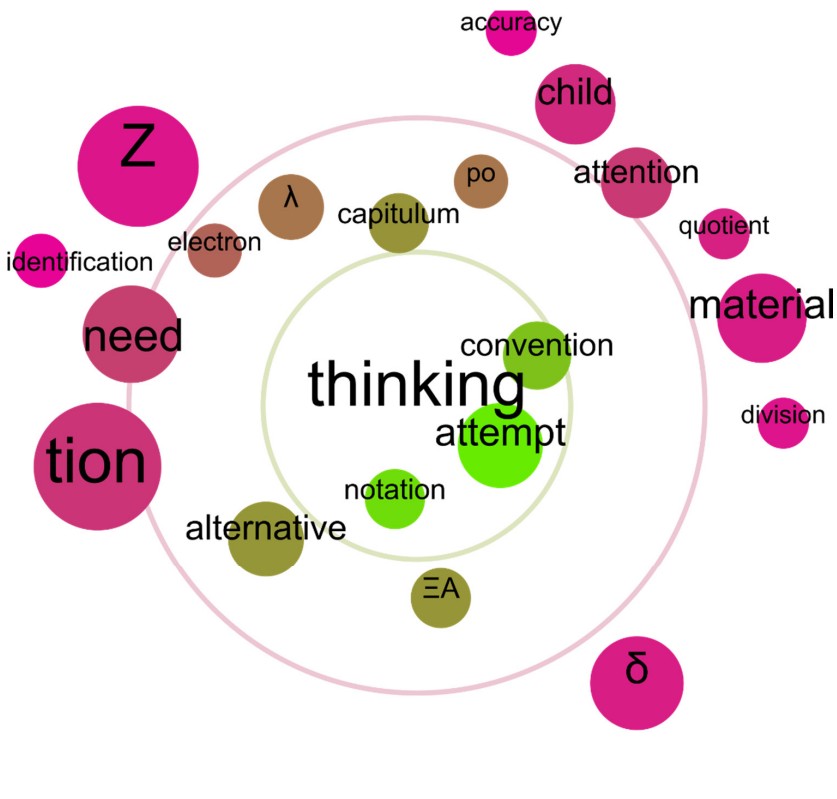

**Figure 4.** Terms semantically close to "thinking" in Turing's corpus.

This semantic similarity is inferred from the resemblance among words embedding the terms of interest. To take a simple example, suppose a given corpus contains two statements:

1. Correlation does not imply causality.
2. Correlation does not imply causation.

The software then should infer that "causality" and "causation" are semantically similar because they are embedded in similar statements, comprising similar terms in similar grammatical structures. Of course, the size of the corpus increases the reliability of such inference, but even for relatively small samples, the results can be useful for further interpretation.

The thesaurus in SketchEngine is generated in two steps. First, the corpus is processed to extract specific expressions (for example, objects of a given verb, such as "imply"), using surface grammar specification in terms of part-of-speech tags. These tags are assigned to word forms at the time of corpus initialization (along with lemmas, or base forms of words). Next, similarities between extracted expressions are computed by comparing collocations that bear the form of these specific expressions. If two expressions share a fair number of collocations, they are considered close in meaning.[7]

Let us then study terms that are semantically close to "intelligence" first.

Now, let us complement this with a grammatical variant, "intelligent", in Figure 2.

Before we draw any conclusions, let us now compare this with semantic fields of "mind" (Figure 3) and "thinking" (Figure 4).

What seems evident is that Turing's terms "intelligence" and "intelligent" are not related explicitly to "thinking" or "mind". By looking at Figures 1 and 2, one can spot connections among various terms. The semantically closest term to "intelligent" is "universal", which is symbolized by the location of the circle containing "universal" very close to the center of Figure 1. The concordance for "universal" (i.e., all occurrences of the term "universal" in the corpus) shows that Turing wrote frequently of universal machines (42 occurrences in the corpus) and universal digital computers. Similarly, he used the analogous expression "intelligent machine" (2 occurrences) and "intelligent machinery" (overall, the term "intelligent" was used 25 times). What Figure 1 also depicts is the fact that the term "unorganized" was applied similarly to machines. Figure 5 shows modifiers of the term "machine" in Turing corpus. While textual connections between these terms are typical of synonymy, it should be stressed that because of the size of the corpus, semantic relationships depicted on Figures 1–4 are best understood as similarity between complex expressions that embed the terms of interest.

Thus, relationships between intelligence (Figure 2) and randomness, making mistakes, and unorganized machines (this is what we saw in Figure 1 as well) are of similar nature: They reflect similar word embeddings. However, all these relationships also suggest that Turing hypothesized that learning was crucial to intelligence, particularly in his *Mind* paper, and that learning can proceed efficiently in a (partially) random fashion. In contrast, the term "mind" is more related to "knowledge", but also to "test" (as can be gleaned in the corpus concordance, the test in question is the imitation game test). Thus, there is a latent connection between having a mind and being intelligent. This is because the imitation game is supposed to play a decisive role in substantiating claims about machine intelligence.

Interestingly, for Turing, the terms "brain" and "mathematician" are equally close to "mind", but the term "child" is closer. This suggests that even though Turing is best known to have analyzed the human mind as performing calculations (in his famous paper that introduced the concept of the universal machine), he speculated about the child's mind development on many occasions, in particular in similar contexts. Turning to closer reading, one can see that by looking at the cognitive development of a child, Turing mused, one could discover the principles that organize our cognitive machinery ([15], pp. 460–463). Even though he would say that it is inessential that the brain "has the consistency of cold porridge" ([15], p. 477), and although the brain is all but mentioned in his 1950 essay that introduced the idea of the Turing test, he did consider the issue of how the brain contributes to the mind as well. This can be easily seen if we compare the adjectival collocations for

"brain" and "mind" (see Figure 6). He would also use the terms "electronic brain" and "mechanical brain" (but critically, see ([15], p. 484)). These terms were made popular in the British press after the war (see [49]).

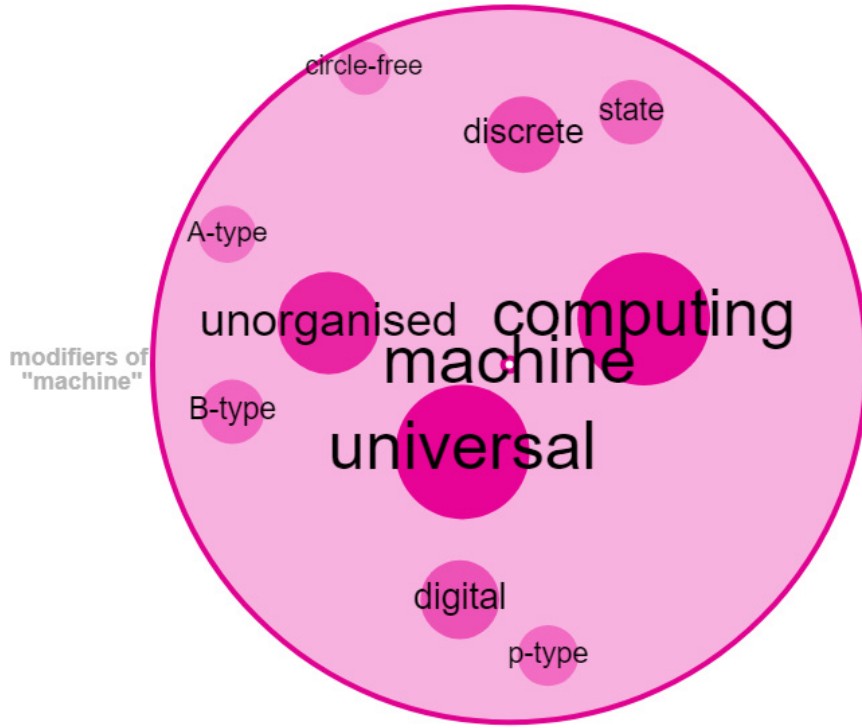

**Figure 5.** Modifiers of the term "machine" in the Turing corpus. To create this diagram in SketchEngine, click "Word sketch", enter the term "machine" and click GO, and then select the column showing *modifiers of "machine"* by clicking the icon "Only keep this column . . . ". Next, click Show Visualization on the top right-hand side. The circle sizes reflect frequency, while distance reflects the collocation strength.

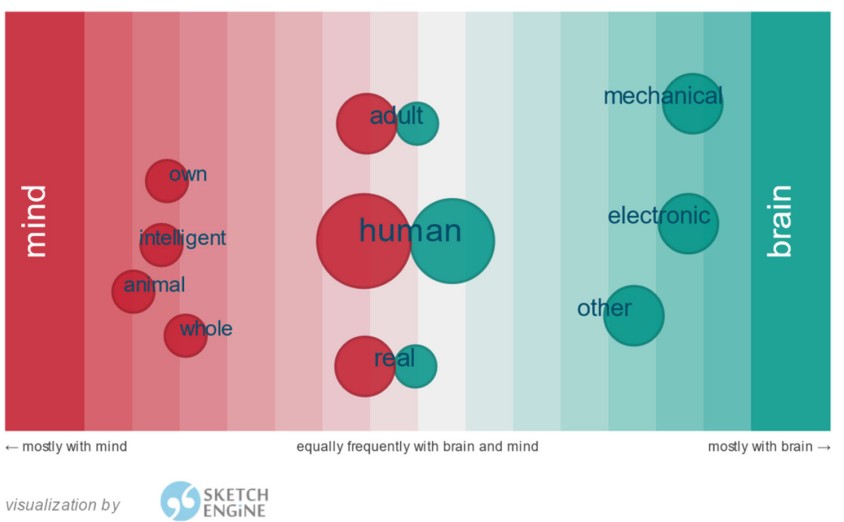

**Figure 6.** A comparison of adjectival predicates that accompany the terms "mind" and "brain". To the left, we find the terms associated mostly with "mind", and to the right, those associated with "brain".

However, all this does not yet answer our initial question: why did Turing talk of intelligence and not mind, when considering computing machinery? What the corpus analyses show is that the notion of intelligence is much more technical for Turing than the notion of "mind", which is used mostly in a nontechnical fashion, without any attempts to sharpen it. "Intelligence" is also related to notions such as "learning" or "memory". This may suggest that his choice was to use a term that was less entrenched in the philosophical tradition of the mind–body problem. Even though "the mind" has a relatively recent origin in modern philosophy, introduced by John Locke in English, but still unavailable in direct translation in many other languages (such as French or German), it immediately raises philosophical concerns, such as "asking what could be known with certainty even if the empirical case was granted" (to cite Allen Newell commenting on the philosophical reception of Turing's 1950 paper ([50], p. 47)), instead of inspiring further theoretical and practical exploration. As many of his contemporaries, such as Ryle and MacKay, Turing preferred to focus on intelligent action.

The semantic relationships inherent in the corpus suggest that Turing's conceptual engineering of "machine intelligence" indeed proceeded in three steps. First, the initial question was replaced with a test scenario. Second, the issue in the scenario was to judge the conversational behavior, which implicitly related "intelligence" to the extremely flexible use of words. These two steps circumvented Wittgenstein's critique by replacing the question with a Wittgenstein-like scenario. Third, Turing offered rebuttals to counterarguments and linked his scenario to the topic of intelligent machinery and possible routes towards achieving it (such as learning). This is reflected in indirect links between the notion of thinking, intelligence, and computation. As Figure 7 indicates, "thinking" for Turing is indeed very close to "computation".

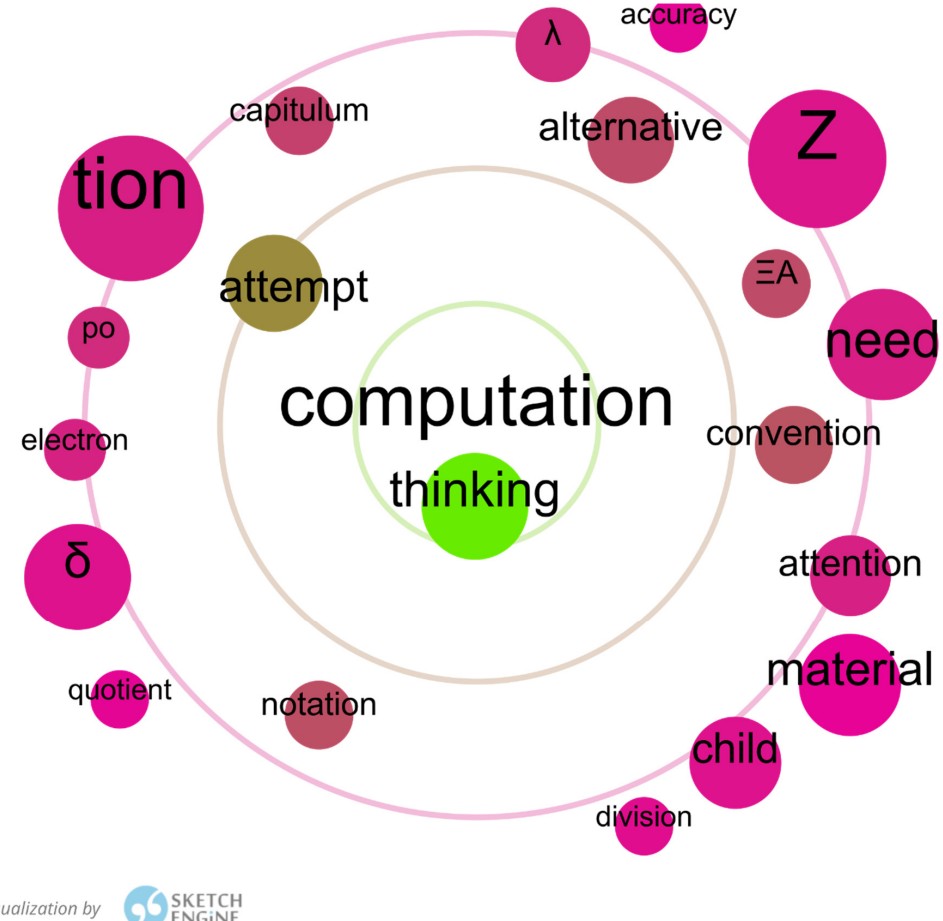

**Figure 7.** Terms semantically similar to "computation" in Turing's corpus.

Surely, language technology does not offer groundbreaking new insight into Turing's choice of terminology. However, it does show that the usages of terms such as "mind" and "intelligence" in his writing are different, but not *vastly* different (for example, the child's mind is mentioned as a possible target of simulation in his 1950 paper). This means that it is wrong to assume that Turing makes a clear-cut distinction between "mind" and "intelligence" (they must be *somehow* related if his conceptual engineering retains the same topic). All in all, this seems to suggest that Turing would probably join his contemporaries in using the term "artificial intelligence" in 1956, rather than "artificial mind", although he would not make a big fuss about it. Both notions are interconnected in his work. This interconnection is responsible for what might be seen as a mild inconsistency in deeming the question "Can machines think" meaningless and answering it. However, if the topic of inquiry is to be retained, the original question cannot be entirely rejected. It is replaced for clarification purposes, but there is still a deeper conceptual connection.

Let us now turn to the issue of why Turing did not treat his imitation game as the operational definition of intelligence. After all, as many other operational definitions, Turing's imitation game does suggest a sufficient condition for something to be a *definiens*—an intelligent entity capable of verbal conversation. Let us then see how Turing would use the term "definition" by looking at related terms (see Figure 8).

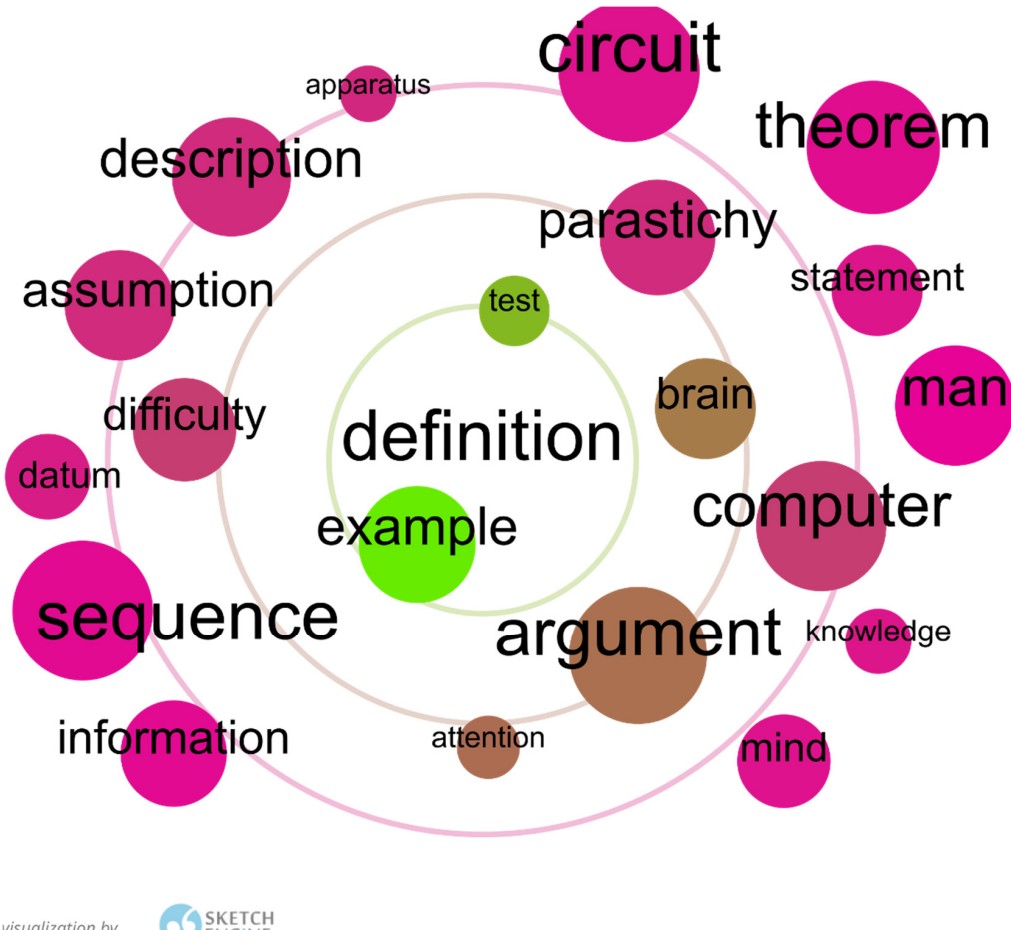

*visualization by* SKETCH ENGINE

**Figure 8.** Terms that are semantically close to "definition" in Turing's corpus.

There are two terms that come closest to "definition": "test" and "example". This shows two things. First, Turing's readers who assumed that his 1950 paper offers a definition by offering a test were not hallucinating. There *is* actual textual evidence that these terms are used in similar situations in Turing's oeuvre. It is obvious why examples play the role of definitions—these are simply demonstrative definitions. Tests can serve

as operational definitions. While the notion of "test" is used much less frequently than that of "example" or "definition" (which is represented by the font size in Figure 8), one can, nonetheless, draw a conclusion that Turing's test is closely related to a definition. A more detailed study of the occurrences of the term indicates that it was used by Turing not only in his famous 1950 paper (and related previous work), but also elsewhere. For example, in his co-authored work with C.W. Wardlaw, a botanist, there is a passage about tests of the theory of morphogenesis proposed by Turing. The passage is worth analyzing in more detail:

> Because of the complexity of all morphogenetic processes, relevant experimental data may be difficult to obtain, but the task should not be regarded as an impossible one. An evident primary test of the theory will consist in the closeness of its applicability to a wide range of biological materials. ( . . . ) Indications of the validity of the theory by the method of prediction have already been obtained by Turing, using the digital computer. ([43], p. 46)

This passage is found in Section 4, "Tests of the theory". Notice two interesting features. For one, testing is here related to the validity of the mathematical model of morphogenesis. For another, Turing provided a computer simulation to make the model plausible, which is not an empirical test of the theory at all. The simulation provided evidence that the model generated appropriate (expected) patterns. This type of testing has become later entrenched in the research on artificial intelligence, meant to shed light on human cognition: the plausibility of results is supposed to indicate the validity of the theory. In a nutshell, this implies that Turing did not constrain the notion of testing so that it would apply only to experimental or observational evidence for or against a theory.

Similar uses appear in Turing's writings about cryptoanalysis, where he speaks of testing potential solutions to the cipher. He also writes about testing the validity of the proofs. This passage is also remarkable:

> ( . . . ) it has been shown that there are machines theoretically possible which will do something very close to thinking. They will, for instance, test the validity of a formal proof in the system of Principia Mathematica. ([15], p. 472)

Thus, testing need not involve experimentation or observation, although it is Turing who discovered that computer simulation may actually involve something akin to experimentation. While working on the Enigma cipher, he needed to perform actual number crunching on machines to test out hypotheses about a particular setting of the German machine used to produce a given message.

To generalize, it seems that, for Turing, testing involves a decision procedure that shows that a certain solution is acceptable. This is what Turing's test is supposed to achieve as well. In this case, his test is also akin to a legal test case that sets a precedent for further conceptual uses.[8] Therefore, while his testing cannot be understood in narrow operationalist terms—concepts are not reduced to sets of experimentational verification operations—it can apply to conceptual structures. It is part and parcel of his conceptual engineering. While Turing denied that he offered a definition, his readers could understand his test as a kind of working definition. However, what they missed is that Turing's proposal was not to replace all theoretical understanding with a simple test operation. On the contrary, the test was to legitimize future work on a thoroughly mechanized theory of intelligence.

## 4. Conclusions

My aim was to present Turing's conceptual engineering in two ways: by looking at it through the lens of traditional close reading and by complementing this reading by additional insights provided by language technology.

The language technology perspective addresses two questions that remained unresolved by the close reading of Turing's work. The first issue is why Turing substituted the notion of mind with that of intelligence. The second issue is whether his test could not be understood as an operational definition that provides a sufficient condition for being

intelligent. While the size of Turing's corpus is fairly limited, which makes any quantitative techniques prone to noise and bias, the analysis suggests some answers to both of these questions. As far as the first one is concerned, the mind/intelligence distinction is not clear-cut, and both terms are interrelated (through the notion of learning or education). Turing preferred the term "intelligence" when proposing developing machine capacities in solving problems and learning. This is because he ultimately wanted to work on a theory of mechanical thinking, clarified in terms of "intelligence." The second issue can be answered simply: although explicitly Turing denied that his test suggests a definition of thinking, implicit semantic relations suggest otherwise. Actually, the Turing test comes very close to being an operational definition. However, the close reading reveals that its role is mostly to motivate the plausibility of the machine intelligence research program by clarifying the notions involved in an intuitive fashion.

The Turing test plays a particular role in how Turing engineers the notion of "intelligence". It is easier to revise notions whose usage patterns are less entrenched in everyday speech and previous theorizing. This makes "intelligence" a much better candidate for conceptual engineering than "mind". The test's role is to change our outlook on intelligence in general, so that we could develop a better conceptual and theoretical grasp of what it involves. However, the test does not offer a rich theoretical understanding, in contrast to what Turing achieved with respect to the notion of "computation". Given the high standards he had for definitions, one can suspect he would be much more impressed by a theoretical definition that provides more rigorous insight into what intelligence involves. Conceptual engineering must start from what is already available, and the grasp of the notion of intelligence in the 1940s and 1950s was insufficient, by Turing's standards, to offer a full-fledged theory.

Looking back, it seems that Turing's attempt to revise our notion of intelligence was a partial success. Indeed, Wittgenstein's cause seems lost: Most English speakers do not consider talking of machines as intelligent, thinking, perceiving, etc., as "somehow nonsensical". For some readers, it did open a rich avenue of theoretical and practical exploration, leading to the systematic study of what was soon called "artificial intelligence". For others, it is as nonconclusive as any other conceptual argument in the debate on the nature of mind as opposed to machine, mechanism vs. biological agency, and so on. However, this is the case for most intuition pumps in philosophy; these do not settle philosophical issues once and for all.

Let me finish with a modest remark. From a meta-theoretical perspective, this paper also shows that, while methods of "distant reading" do not offer any silver bullet for difficult philosophical problems, language technology can still provide us with more understanding of our intellectual heritage. Nonetheless, while language technology is useful for qualitative analysis, its reliability increases with the quantity of text.

Needless to say, there are serious technical obstacles for computational methods in the service of intellectual history. For example, while a huge number of historical materials were scanned and made publicly available, their availability, even for research purposes, is still insufficient. For example, the British Newspaper Archive offers merely a search engine, but bulk downloads, detailed textual analysis, and similar functions are not accessible for copyright reasons. Moreover, handwritten manuscripts, typed correspondence and notes all remain very difficult targets for optical character recognition; instead, they usually must be manually transcribed, which makes the cost of such operations fairly prohibitive at a larger scale. Having a corpus of all Turing's writing, historical newspapers, technical reports, and academic writing in related disciplines, could provide more insight into his work. It is not unreasonable to expect that these resources will become available in the future (Turing's heritage will enter the public domain in 2025).

Hopefully, Turing would be pleased that algorithms can offer some insights into his own thinking. While these would not pass his test, they bridge the gap between the humanities and artificial intelligence. It is still the job of the human being to interpret the result of the analysis.

**Funding:** This research received no external funding.

**Conflicts of Interest:** The author declares no conflict of interest.

## Notes

1    Some critics of ordinary language philosophy ascribe this fallacy to this research tradition as a whole [5]. However, this seems to be insufficiently charitable. Mere frequency or typicality of usage is not sufficient for ordinary language philosophers to justify its correctness. In fact, ordinary language philosophers were much more sophisticated in justifying their normative claims, for example, about category mistakes involved in typical usage patterns (see, e.g., [6]).

2    In this, Turing remains close to ordinary language philosophers, such as the late Wittgenstein [10]. Remember that Shanker ([9], p. 2) opines that Turing offers a Wittgenstein-like argument.

3    A number of critics, starting from Claude Shannon and John McCarthy ([15], p. 437), through Stanisław Lem [19], to Ned Block [20], objected that the process could be quite "nonintelligent", while still producing intelligent-like conversations. The plausibility of the idea that a simple "lookup table" is capable of producing and understanding a potentially infinite number of English expressions over 30 min of unconstrained conversation is debatable, in particular because these lookup tables are not supposed to track the history of the ongoing conversation. Our current deep neural networks with hundreds of billions of parameters display remarkably fluent linguistic behavior, but remain brittle for common-sense questions and still fail to track the history of interaction.

4    Turing speculated that a machine would pass his test in a hundred years from 1952 ([15], p. 452), so the situation may look different in thirty years' time. Note that he mentioned the end of the 20th century in his 1950 paper as the time when "the use of words and general educated opinion will have altered so much that one will be able to speak of machines thinking without expecting to be contradicted" ([15], p. 449). This latter claim does not imply that the judges at the turn of the 21st century would consider any machine capable of imitating human conversation, however.

5    Unfortunately, the official website of the Cambridge Turing Archive is down as of 17 January 2022, previously available at www.turingarchive.org, (accessed on 10 June 2022). This makes crucial Turing online archives no longer accessible. Unfortunately, these also did not include full transcripts of Turing's correspondence (OCR for typewritten or handwritten text remains very noisy), such as those found in 2017. For this reason, the corpus does not cover all remaining correspondence of Alan Turing. The corpus does not contain Turing's writings in pure mathematics [44] and his unpublished work in logic from [45] either. There are two reasons for this exclusion: (1) the optical character recognition (OCR) of mathematical notation is extremely noisy; (2) collocation analysis does not produce meaningful results for English documents that are predominantly mathematical. While the writings in [15] were already digitalized and proofread, other work had to go through OCR. For this purpose, tesseract OCR engine was used (in the default LSTM setting). A more detailed description of the corpus along with all modeling results is found in the repository RepOD available at https://doi.org/10.18150/WSTA4E (accessed 15 June 2022).

6    To recreate the diagrams, open the Turing corpus in SketchEngine, and then click the Thesaurus Table on the Basic search tab, enter the term of choice, and click GO. When the textual table appears, click Show Visualization. All output data produced this way is available in the public repository, including the logDice metric of collocation strength and frequency of the related term.

7    In this respect, SketchEngine differs from typical collocation extraction methods, including word2vec models, which are oblivious to the grammatical form of underlying expressions. The association score used to extract collocations in SketchEngine is relatively corpus-independent [47], in contrast to other association scores used in collocation research. For example, Alfano's method does not involve any surface grammar constructs and relies on term frequency and co-occurrence [48], but these are comparable only in a single corpus. In general, collocation research often relies on variants of mutual information score [41]. This implies that semantic similarity in the corpus under study may lead to somewhat diverging results if other association scores are assumed. It must be stressed that various language technologies may serve different goals, and provide other insights into the text. For example, a word embedding model built using the word2vec algorithm [38] from the Turing corpus does not provide the same results as SketchEngine because its reliability requires much more data (e.g., the terms closest to intelligence in this model are "near", "during", "bury", "irregularities", "early", "supply", and "discrete", which are far from informative). The word2vec model is available in the repository.

8    I owe this observation to Joanna Loeb.

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
