# Peer review of "Turing’s Conceptual Engineering"

_philosophies, doi:10.3390/philosophies7030069_

Round 1
Reviewer 1 Report
This manuscript presents the author's answer to two questions raised by Alan Turing's work: (1) why did Turing use the word intelligence instead of mind? (2) what is the Turing test? (is it an operational definition or not?)
In addition, the manuscript presents Turing's work as a form of conceptual engineering, specifically of the concept of intelligence.
In addition, the manuscript uses a combined methodology of close reading + language analysis to show why the author's interpretation is justified.
Each of the three goals of the paper are separate and could have been each a paper in itself, and do not inform each other, although the author claims that the specific method of analysis does give us unprecedented insights.
Speaking directly to the author, hi, how are you? I had a lot of difficulties in understanding your text. I couldn't follow the argument. You need to take pity on your readers and explain to them what you are doing and why. Take us, readers, by the hand and explain to us each step of the way. Because, if you talk to yourself without any care for what others might understand from this, you will lose your audience on page 1. I read your manuscript because I had to review it, but if I had to encounter it in a journal, I wouldn't have gotten past page 1. If you want to make a difference in the scholarship, please think of your audience.
Here are some recommendations to make your argument stronger and clearer.
- You need to not assume that your audience knows stuff about Alan Turing, linguistics, Turing tests, conceptual engineering. So you need to explain each technical term when you introduce it. Not only because your audience may not know these terms, but also because we may have other definitions in mind than you.
- make clear for yourself what is your main point here? Is it to show that Turing was doing conceptual engineering without him knowing? Is it to answer the two questions? is it to show how awesome language analysis is for philosophers? Once you choose one (or all points), cut anything that does not connect to this point. You have a lot of distracting mentions in the document. a lot of mentions of things that have nothing to do with your claims. Example on page 2 " the symbol grounding problem 79 (SGP), formulated by Stevan Harnad [8], is a pseudoproblem" Why are you telling us this if you do not explain in detail what this problem is and why it is a pseudoproblem? It is distracting and does nothing for your argument. It looks like theory-name dropping. These kinds of detours are filling the paper and make it very hard to read. On pages 3-4 you deconstruct a paragraph from Turing's work, translate it into logical arguments, and then show us it is a reductio ad absurdum. why do you do this? First, the reconstruction of the paragraph is not faithful, and secondly, it is not a reductio ad absurdum just because Turing says something is absurd. And do you need this (incorrect) logical deconstruction to make a point? It is distracting and I did not see its point there.
- stop putting words in people's mouths. This is the most disturbing feature of your paper, and the most unscientific. You cannot reconstruct what people say by interpreting what they say to fit your concepts. You do this first with Firth on pages 2 and 3, and then, for the rest of the paper, with Turing. For example, on page 2 "His (Firth's) assumption seems to have been that meaning is reducible to intertextual relations between words". But did he say that or do you interpret it to be so? Because if he did not say it, you are attributing to Firth radical views that he maybe did not hold - and this is not ok, research ethics-wise. If he did say it, then it is not an assumption, but a premise. You need to be clear on the terminology here. And to provide quotes to back up your claims.
- Related to point 3, i would advise against using normative language to characterise what various authors do. Firth is radical, philosophical conceptual analysis is "simplistic", some interpretations are innocuous, etc.
- You need to contextualise properly this paper. Are the questions you are asking (namely 1 and 2) asked before by others? if yes, how do they answer? If not, why not? It seems like you read Turing, found some interesting questions -for you - and then showed us how you would answer them. But why should we care about 1 and 2? You need to embed this paper in a wider literature on the philosophy of Turing in the introduction already. We should not wait to get to page 5 to find out with whom Turing may have been at odds. You mention Fodor, behaviourism, functionalism. Why do these debates matter here?
- I would really downsize section 2 on linguistic analysis as a method. It contains a lot of bold claims of why language analysis is more awesome than close reading, and it gives us some hints at some debates in linguistics. Why are you telling us all these fights between Firth and the competitors? Is it useful for the argument on Turing? You introduce a lot of technical terms, I do not know to which one to pay attention, and by the time you reach section 4, I have forgotten all of them. You can move the explanation of the method closer to section 4, tell us how close reading is failing us after you raised some questions about Turing's work, and then bring this method as additional insight. But as you present it now, in inflated terms, this looks like THE method we should all follow and I still do not understand why some debates among linguists should have an influence on how we do philosophy.
- every argument where you mention conceptual engineering is weak because you never explain what you understand by conceptual engineering. this is the weakest point of the paper, which breaks its spine. You give some references about conceptual engineering, sure, but do not explain what it is except that it is of two kinds (definitional and revisionist). OK, but to your mind, what is conceptual engineering? How is a concept different from a definition or from the meaning of a word? At some point, you talk about an intuition pump and how it is supposed to solve philosophical problems. This made me realise that all philosophical problems may be conceptual problems, in your reading. I am not sure about that, but, just as Turing says that you cannot solve the problem of can machines think by analysing the definitions of machines and thinking. So conceptual analysis only gets us this far, we need to do more (in philosophy). So when is conceptual engineering called for, in your interpretation? I would really drop the whole line of argument "Turing was doing conceptual engineering before it was invented" if you do not explain what that is.
- In the quote you analyse from Turing, page 3, he talks about definitions. Why do you reconstruct it as lexicographic definitions? Turing never mentions dictionaries... You seem to put words into his mouth without justification.
- in your answer to questions (1) and (2), you contradict Turing. This is very strong. He says he did not want to define intelligence, you say it was an operational definition. OK. You need to define what is an operational definition and why this matters to us. You also need to draw the conclusions for scholarship from your answers to 1 and 2. Again, why should we care about these answers? If you merely provide us with insights, as claimed in the introduction, that is a very weak claim. what should we do with these insights?
- There are many many more contradictory or unexplained claims in the paper. If you do not need it for your argument, just drop a sentence. If you need it, make sure you explain in detail what it is.
- Also I would advise against name-dropping and theory-dropping. You have these one-liners where you hint at a debate, such as the sybmbol-grounding problem, but many many more, you solve it in one sentence and then move on. This is very confusing for the reader. Why are you telling us these things?
- You need to explain to readers how to read the images from your language analysis. what stands for what. You need to clearly demarcate what the language analysis show us exactly, but in simpler terms, without the linguistic jargon. It's confusing.
Author Response
Please find my reply attached.

Reviewer 2 Report
Short summary of the report (for the whole report, please read the attached pdf-file):
The article aims at re-interpreting Turing’s work on intelligence as an instance of conceptual engineering, thanks to a mixture of traditional philosophical analysis and algorithmic techniques for language processing. The article is well-written and clearly structured, the proposal is interesting and (for the most part) well-argued. Moreover, the combination of classical philosophical methods and algorithmic techniques is original and insightful.
There are, however, some issues of philosophical clarity and scope that preclude me from recommending the paper for publication, as it stands. My main issue with the paper is that the notion of conceptual engineering that the author attributes to Turing is quite vague and it does not really explain what exactly changed conceptually with Turing’s seminal paper in the scientific/philosophical notion of intelligence. This issue stems from some vague statements in the argumentation and from a lack of situation of the paper in the relevant philosophical literature. In order to remedy these infelicities, the author should explain better what his/her/their proposal amounts to, engaging more with relevant existing work about Turing’s philosophical methodology and Turing’s conceptual engineering
on computability.

Author Response
Please find my response attached.

Reviewer 3 Report
This paper applies an interesting digital humanities methodology to explore Turing's project in Computing Machinery and Intelligence. The paper argues that Turing should be thought of as engaging in "conceptual engineering" on the concept of intelligence. This is initially developed using the traditional methods of the history of philosophy (close reading). This close reading is backed up and deepened by a "distant reading" method analysing a corpus of Turing's writings. The author argues, using this distant reading, that there is a loose sense in which Turing's work provides definitions of key terms and traces out some interactions and connections in Turing's use of "thinking", "mind", "intelligence", "brain" and "definition".
I hope that the following comments will help the author in any further development of this interesting research project. I'll first set out the overall concerns I have and then give a few specific comments.
Structure: The introduction does not give a sufficiently clear sense of what will be argued for and how it will be argued for. For instance, the paragraph starting line 31 begins to set out what will be discussed section-by-section but this is not completed. Throughout, it was hard to know what claims the author is arguing for - at some points it seems like they're arguing that Turing does not introduce a definition of intelligence and that the distinction between talking about "mind" and "intelligence" is important but then they seem to walk these claims back towards the end of the paper. It would be much easier to understand the structure of the paper if the author clearly stated at the beginning what their eventual conclusions will be and how they intend to reach them. This is especially true for an innovative methodology of the sort employed in this paper
Section Three is titled "Why not common-sense intuition?", but discusses much more than this. This section should be retitled and divided into parts with more obvious functional roles.
Close reading: My primary concern in the close reading sections is that it is not clear exactly what the author means by conceptual engineering and what the relationship between conceptual engineering and intuition pumps is. The language of 'intuition pumps' seems to me to be used in the context of conceptual analysis just as much as in conceptual engineering.
A more fundamental point is that, if Turing is being presented as a conceptual engineer, it should be clear (a) what was the notion of intelligence before Turing, (b) what was wrong with it, and (c) how did Turing change it? I can see gestures towards all three in the paper but would like to see them set out much more clearly. As an example of unclarity about this: line 319 talks about Turing making changes in the notion of intelligence possible as opposed to straight forwardly changing the notion of intelligence.
In the argument reconstruction beginning on line 137, I do not see the conclusion to the argument. The argument would be much clearer if it included the conclusion. I would also like to see further evidence that Turing entertained your premises 1 and 2. The idea of a 'conceptual question' isn't explicitly used by Turing in the paper and while it is clear in the quote you give that Turing thinks common usage is a way that definition might be framed, it is a long was from this to premise 2. There are surely other ways to set up lexical definitions.
I do like the connection the author makes between the rejection of statistical survey's and the broader idea of conceptual engineering. There is good material here.
The author can take or leave my suggestion that they engage with work on Charles Peirce regarding the role of definitions in philosophy. The second and third chapters of Christopher Hookway's Truth, Rationality and Pragmatism (OUP, 2002) argue that Peirce does not give a definition of 'truth' but a 'clarification' of the practical import of claiming that something is true. This example might help to bring out what you mean by a 'novel understanding' of intelligence which is not a definition.
Distant reading: I believe the author spends too much time on the philosophical underpinnings of distributional semantics. As they rightly note towards the end of the section on distributional semantics, the methods of corpus analysis can have heuristic value even if distributional semantics doesn't have a good answer to the "symbol grounding problem". There's no need to get lost in the weeds here and much more beneficial uses of the space to bolster the author's main arguments.
Turning to the actual methods used, I think it is important that this kind of work adopts the principles of open and reproducible research. Ideally, the author provides supplementary material which allows a complete reproduction of all technical steps to produce the analysis and plots provided. While it would be unfair to require this of the author, given that it is not yet a clear expectation in the discipline, more moves in this direction are required.
For instance, there is some discussion of what goes in to the corpus but more detail is necessary. Often this can be done using a table which, in this case, should at least describe the source of all the text, how much text comes from each source, whether the text is generated by OCR or not, and perhaps what the topic of the text is. Are coauthored texts included? One is cited, but it is not clear it is in the corpus. It is especially important for this project to add the main genre of the text, given that the use of 'definition' etc can be more technical in mathematics than in philosophical discussion.
More detail is also required at the analysis stage. The author uses the online and commercial service SketchEngine and says that the details of the algorithm are proprietary. This is unfortunate given the widespread availability of open source tools for generating this kind of analysis. That said, it does allow for greater accessibility to researchers without a background in coding. My primary complaint here is that the author does not provide further detail into the settings used with SketchEngine so that another user of the programme could recreate something like what the author has done. Secondarily, while the details of the algorithm might be propriety, a cursory investigation of the SketchEngine website reveals that they do provide some detail about what they do. Some of this should be conveyed to the reader (or at least cited in more detail).
It is not clear to me how to read Figure 1. At a minimum it would be necessary to explain what the different cells in the figure mean, what it means for a cell to be within another cell, what the numbers in parentheses are, and where the words inside the cells come from. Are they some kind of stopwordless n-gram? It is not clear.
In general, more words are needed to make clear what the Figures are supposed to convey. At line 468-469 it is claimed that the connection between universal computing machinery and intelligence can be "easily" spotted. I have a sense of what the author means, but this needs to be spelled out.
At line 480, the figures generated by SketchEngine are used to conclude that Turing took the 'child's mind to be even more important' than 'the human mind as performing calculations'. It is hard it see how proximity of terms in these figures can do the work required to draw this conclusion. It would be unobjectionable to use these figures to raise the question of why 'child' is related to 'mind' in Turing's writings and to use this as a lens for returning to close reading. They cannot be used, it seems to me, as a direct test for what Turing took to be 'important'.
The use of Figure 6 worried me. The fact that 'adult', 'human', and 'real' appear (I assume) immediately to the left of 'mind' and 'brain' with roughly equal likelihood (however this is being measured) is not sufficient to show that 'Turing uses the terms "mind" and "brain" interchangeably when talking about adults and human beings'. If I wrote something where I carefully distinguished between the "human mind" and the "human brain" you could very well end up with the same kind of result as in Figure 6.
Finally, for this section, note that the term "distant reading" is closely associated with Franco Moretti and he should be cited (perhaps his collection Distant Reading, and especially the text "Conjectures on World Literature" within the collection.
Interaction between close and distant reading: The interplay between the two methodologies the author uses is not clear. First, the motivation for turning to distant reading is insufficient. On Line 376 it is claimed that two key questions are 'unanswered' by close reading, but there is no evidence provided that they are not answerable by close reading or that there is any reason to prefer distant reading.
In addition, in my comments above I have complained about the use of distant reading methods to draw certain conclusions. I think the problems I raised there are best dealt with by considering the distinct roles of the close and distant reading methodologies. Any future version of this paper should clarify exactly what is being argued and what aspects of the overall argument are provided by the close and distant methodologies.
One thought which occurs to me is that term networks of the sort presented in this paper could be used as proxies for changes in concepts over time or over different texts. If we want to show, for instance, that Turing engineered the concept of intelligence, we might want to look at the networks for 'intelligence' in Turing and before and after Turing in some suitable comparison corpus. This kind of distant reading methodology would have a more obvious connection to the author's research questions.
I think engagement with part one of Mark Alfano's Nietzsche's Moral Psychology (CUP, 2019) would help here. This chapter sets up a similar methodology to the one employed in this paper and could provide a good example for future developments of the research project.
Additional comments: A citation method which includes page references should be used for this kind of project. It is important that the reader can find exact passages. For instance, at line 66 much more detail is required for the reader to chase up the passage from Firth. It's not enough to point the reader a volume of selected papers, we should be given the title of the item within the collection and a page number.
Lines 200 and 369 seem contradictory: one claims Turing thinks conversational ability is sufficient for intelligence the other seems to says he doesn't.
Line 212: for clarify it'd avoid using the phrase 'machine settings' here, which suggests the settings of the machine to be tested rather than the rules of the imitation game that Turing proposes.
Line 297, footnote 3. Minor point of fact: Turing claims 50 years in "Computing Machinery and Intelligence" rather than 100 years. Perhaps I am missing a distinct prediction Turing made. If so, this just proves the earlier point that more specific citations are needed!
Line 557: I wasn't clear why the idea that simulations count as experimental evidence should be surprising here.
A trivial point: I believe the author means 'Figure 5' at line 496 (rather than 4).
I hope these comments are found to be helpful.
Author Response
Please find my reply attached.

Reviewer 4 Report
Report on
“Turing’s conceptual engineering”
submitted to Philosophies
The stated purpose of the article is to argue/show that the celebrated “Turing test” is actually a piece of conceptual engineering. For what it is worth, I find that very plausible, mostly in light of what Turing himself says in that article. What would be needed for this is a discussion of what conceptual engineering is, and how it differs from other philosophical enterprises, presumably something like conceptual analysis. There is very little of this in the present paper.
Following this plan, I would be interested to know what the author thinks of Tarski’s account of logical consequence in the 30’s. Like Turing, Tarski also says he is not concerned to capture the everyday notion. The author describes both Turing’s 1950 paper and his classic one on computability as sharpening a previous, everyday concept. Is the claim that every proposed sharpening is an instance of conceptual engineering? Is that all it takes?
As I understand it, conceptual engineering is a proposal to replace a concept in use with a different one, arguing that it will serve some purposes better. If this is what is going on, then what are the purposes for which the Turing test is proposed?
And I failed to understand the point of the detailed analysis of Turing’s writing, to show why he was interested in “intelligence” instead of “mind”. Perhaps that material is of some interest, but it is not presented very clearly, nor do I see how it bears on the matter of conceptual engineering.
I also did not follow the arguments over whether the Turing test is an “operational definition” or not. Again, perhaps that is of some interest, but I don’t see what that is, nor do I see how it might bear on the matter of conceptual engineering. It might help to be told what an operational definition is, why it matters (and what it matters for), and how it bears on the stated theme of the paper.
On p. 2, we are told of “distributional semantics”, but not what that is, nor (again) how it relates to the matters at hand. The rest of that page and most of the next is rather unclear. How does natural language processing relate to a claim that a historical figure is engaged in conceptual engineering?
Also on p. 2, “The resultant radical feature of this approach to this approach to meaning is that it seems to have no place for reference.” Why? How? And, again, why is that even relevant?
I have no idea what a “distant reading” is, nor about how that helps the present case.
On p. 4, I could not follow the reasoning in the second paragraph. The author has Turing saying we should not confuse “is” and “ought”. Good advice. But how is that relevant here? If he was engaged in conceptual engineering, then he would be saying that we ought use the term “intelligence” in such and such a way. For what purpose?
Also on p. 4, we are told that the “core idea is that there could be sufficient similarity of how machines could declare their thoughts to others, as compared to human beings.” Doesn’t that beg the question, presupposing that machines have thoughts?
At the top of p. 5, we are told that Turing and Descartes took language use to be sufficient for intelligence. This sounds like old fashioned conceptual analysis. How does that bear on conceptual engineering? I am not saying that the case here cannot be made, but it does have to be made.
What is “weakly equivalent”? Is the reader supposed to be aware of this?
On p. 5, we are told that the Turing test “opens up a conceptual possibility that machines could display intelligence”. Can we open up new conceptual possibilities just be redefining our terms (via conceptual engineering)? Seems more like a change of subject to me. (This is a common theme in the literature on conceptual engineering.)
On p. 5, we are told that the test “proved its usefulness as a tool of conceptual engineering”. How? This goes way too fast. Useful for what?
On p, 7, we are told that the Church-Turing thesis “cannot be defended as true (or rejected as false)”. This is controversial. There is a cottage industry of claims that the thesis can be proved (or refuted).
To repeat a point I made earlier, on p. 7, we find that there are “two questions that remain unclear in light of what has already been said. First, why did Turing stress the notion of “intelligence” rather than ‘thought’, ‘thinking’ or ‘the mind’?” I agree that those questions remain unclear, but don’t see why they are important.
And, again, I do not see the relevance of the entire second half of this article.
Author Response
Please find my reply attached.

Round 2
Reviewer 2 Report
I am very pleased with the author's replies and revisions. All the worries/issues that I raised in my referee report were either addressed in the replies or adequately implemented in the revised manuscript. The revised manuscript, thanks also to the suggestions of the other reviews, is now far clearer in both its aims and methods and better situated in the relevant literature.
Reviewer 3 Report
I am very impressed by the reformulation of this paper. The current version is a pleasure to read. The concerns I set out in my original review are answered adequately in either the cover letter or the reformulated paper. My primary concerns (the methodological connection between the close and distant reading aspects of the paper and the extent to which the distant reading aspects are explained and made reproducible), are dealt with very well.
My apologies for blaming the author for the official citation policy of Philosophies (which, in any case, I should have looked up for myself)!
The author rightly notes that they made the structure of their argument reconstruction (now starting at line 84) quite clear. I struggle to reconstruct how I came to misinterpret it (possibly the closeness of the bar to the end of the page).
Some minor points:
- Line 315: 'any quality metric' reads strangely to me. Perhaps it's too close to 'qualitative metric'. I assume 'a metric of high quality' is meant. Rephrasing this would help clarity.
- Line 327: should 'the definition' be 'a definition'?
- Line 329: I'd add 'but was' before 'sharpened'. It looks like the sharpening is being contrasted with the previous state of the concept.
- Line 387: 'Not only he could not' -> 'Not only could he not' and move 'also' before 'lacking'.
- Line 393: Two spaces before 'in Brit-'? If so, there should only be one.
-Line 452: add 'to' before 'rely'.
- Line 492: 'among words embedding the terms of interest' is still a bit unclear. As a whole, the example which follows is enough to make this section of the paper clear. But this sentence is a bit hard for me to understand. I think it is the use of 'words' and 'terms' in the same sentence without a clear distinction between the two. In the sentence, I would specify exactly what the semantic similarity is between ('words' or 'terms').
-Line 485: Use of quote marks around 'intelligence' would match the rest of the paragraph (and paper).
- Line 623: Starting the sentence with 'If' rather than 'But' reads more naturally to me. 'But' makes it seem like a contrast is being made with the previous sentence, but it is really being explicated.
- Line 624: I think 'for the purpose of clarification' or 'for clarificatory purposes' would work better here (rather than 'for clarification purposes').
- Line 625: 'an' operational definition rather than 'the'?
I've recommended 'accept in present form'. I hope that my comments about line 492 and 315 above will be taken into account before publication, but they don't quite reach the level of a 'revision'.